# Nationalist Mobilization, Ethno-Religious Contention, and Legal Innovation in a Stateless Nation: Explaining Catalonia's 2009 "Law on Centers of Worship"

Avi Astor





Department of Sociology, Universitat Autònoma de Barcelona, 08193 Bellaterra, Spain; avi.astor@uab.cat

**Abstract:** This article analyzes the development and framing of Catalonia's "Law on Centers of Worship", an innovative law dedicated exclusively to the regulation of religious temples that was passed by the regional Parliament in 2009. The law was a legal novelty in Spain, as well as in Europe, where regulations pertaining to places of worship are typically folded into regional or municipal laws and ordinances dealing with zoning and construction. My analysis highlights how the law aimed not only to address the challenges generated by the proliferation of places of worship serving religious minorities, but also to legally reinforce and symbolically affirm Catalonia's political autonomy and cultural distinctiveness vis-à-vis Spain. I place particular emphasis on how the temporal confluence of heightened nationalist mobilization, on the one hand, and tensions surrounding ethno-religious diversification, on the other, contributed to the development of a legal innovation that integrated the governance of religious diversity within the broader nation-building project. This article's findings illustrate the role of historical timing and conjunctural causality in shaping the dynamic nexus between religion, law, and politics.

**Keywords:** nationalism; religious diversity; governance; Catalonia; Spain

## 1. Introduction

In 2009, Catalonia's Parliament enacted a "Law on Centers of Worship" (LCW). This was the first law dedicated exclusively to the regulation of places of religious gathering in Spain. It was also a novelty in Europe, where regulations pertaining to places of worship are typically folded into regional or municipal laws or ordinances dealing with zoning and construction. The LCW was framed as a pragmatic response to the challenges posed by the proliferation of temples catering to religious minorities and the need for clearer regulations regarding their establishment. It introduced a special license for opening places of worship, a series of conditions that had to be met for obtaining the license, and certain requirements for municipalities to ensure adequate space for the construction of religious buildings in their zoning schemes (Martín García 2012). Prior to its enactment, the LCW elicited fierce criticism from the Catholic Church and conservative political parties, especially the centralist Popular Party (PP). This criticism centered largely on the law's alleged futility, secularist militance, nationalist ideology, and animus toward Catalonia's Catholic heritage, despite the fact that the law included exemptions for many existing Catholic churches.

The enactment of the LCW, as well as the intensity of the criticisms leveled by Catholic leaders and conservative political elites, raise several questions: Why did Catalan legislators elect to chart a new path for religious governance by developing the LCW instead of following the example of other European regions by adding articles or clauses on religious gathering and worship to existing zoning and building regulations? Why did the law elicit such strong criticism despite the unlikelihood of it having any tangible, let alone pernicious, consequences for actual Catholic churches? Answering these questions requires an analysis of the dynamic nexus between religion, law, and politics in Catalonia. I argue that the LCW was not conceived exclusively as a pragmatic solution to the problem of

regulating minority places of worship, but also as an instrument for legally reinforcing and symbolically affirming Catalonia's political autonomy and cultural distinctiveness vis-à-vis Spain. I place particular emphasis on how the temporal confluence of heightened nationalist mobilization, on the one hand, and tensions regarding ethno-religious diversification, on the other, contributed to a legislative innovation that sought to address both issues simultaneously.

## 2. Ethno-Religious Diversification, Nationalist Politics, and Religious Governance in Stateless Nations

In analyzing the Catalan LCW, I draw on the growing body of literature on the interrelations between ethno-religious diversification, nationalist politics, and religious governance in stateless nations. Following Guibernau (2000, p. 990), I use the term "stateless nation" in reference to "nations which, in spite of having their territories included within the boundaries of one or more states maintain a separate sense of national identity generally based upon a common culture, history, attachment to a particular territory and the explicit wish to rule themselves". Recent studies have highlighted how the representation and regulation of religious diversity in stateless nations are commonly bound up with projects of nation-building that are premised on notions of cultural distinctiveness and that aim to promote political autonomy. In some stateless nations, ethno-religious diversification has been framed as a threat to core elements of national identity, values, and traditions, with the effect of sparking defensive appeals to the cultural and religious heritage of the nation. For instance, scholars have shown how political elites and other public actors in Québec have invoked the province's Catholic and secularist heritage or tradition to assert the dominance of the majority within an increasingly diverse religious landscape (Burchardt 2020; Zubrzycki 2012).

However, ethno-religious diversification may also present new political opportunities for stateless nations to affirm their distinctiveness vis-à-vis the larger nation-state contexts in which they are embedded. In Catalonia, for example, challenges related to religious diversity have served as an impetus for the development of governmental agencies, normative frameworks, and legal initiatives that bolster the region's autonomy and claims to distinctiveness vis-à-vis Spain (Astor 2014, 2020; Burchardt 2020; Griera 2016). Scholarship on Québec has likewise shown how general frameworks for regulating religious difference and the underlying principles around which they revolve (i.e., laïcité and reasonable accommodation) reflect efforts to augment political and cultural distinctiveness vis-à-vis Canada (Burchardt 2017, 2020; Dupré 2012; Peker 2017; Zubrzycki 2012, 2013, 2016).

The reframing of religion as national cultural heritage in the face of ethno-religious diversification and the entwinement of religious governance with nationalist politics are by no means exclusive to stateless nations (e.g., Beaman 2013; Hervieu-Léger 2000; Joppke 2018). However, given that stateless nations lack power to regulate their territorial borders or the citizenship and legal status of their inhabitants, cultural matters such as language and religion that are within their competency take on special importance within nationalist projects.

A key theme running through much of the literature on religious diversity and nationalist politics in stateless nations is the role of historical timing and developments, both local and global, in shaping the representation and regulation of religious difference. In order to account for temporal fluctuations of nationalist sentiment and mobilization, and for the impact of such fluctuations on social and political action, Brubaker (1996) has suggested conceptualizing nationness as an event that crystallizes during particular historical moments, rather than a stable and constant social force that operates evenly across time. Doing so does not necessarily preclude the recognition of longstanding socio-historical structures or configurations that give form to the general framing and content of nationalist discourses or policies. For example, the historical predominance of Catholicism in both Catalonia and Spain has hindered it from emerging as a marker of national distinction in Catalonia as it has in Québec (Astor 2020; Burchardt 2020). Nevertheless, the manner in which Catholic heritage (or hostility toward the Catholic Church) has figured into discourses and policies

pertaining to the nation or the regulation of religious diversity in each region has varied over time depending on an array of contextual conditions and circumstances.

Hence, rather than considering Catalan nationalism as a more or less constant force that contributes to particular discourses and laws pertaining to religious governance, I examine the dynamic and contingent relation between nationalist mobilization and ethno-religious diversification in Catalonia. I argue that the temporal confluence of conflicts related to ethno-religious diversification, most notably local controversies over the establishment of mosques, on the one hand, and heightened nationalist mobilization, on the other, contributed to the development of the LCW as a legal innovation that integrated the governance of religious diversity within Catalonia's broader nation-building project. To be clear, my argument is not that conflicts related to ethno-religious diversification were themselves a source of heightened nationalist mobilization or vice versa. As I explain below, the sources of ethno-religious conflict and of nationalist mobilization in Catalonia were distinct and, by and large, independent. Yet, the temporal confluence of these developments is critical for understanding why the LCW was developed as a stand-alone law, its particular framing and content, and the manner in which it was presented to Parliament. The explanation advanced below thus emphasizes the role of conjunctural causation in shaping the complex entanglements of religion, law, and politics (Decoteau 2018; Mahoney 2000; Sewell 2005).

### 3. The Novelty of Regulating Places of Worship in Catalonia and Spain

Given the centrality of Catholicism to Spain's emergence as a nation, Catholic churches, parishes, and chapels have long been core to the built landscape of Spanish towns and cities. Although there is a lengthy history of reflection about the proper religious procedures for consecrating churches, legal and political debates regarding their regulation did not surface in Spain until the end of the 20th century (Rodríguez Blanco 2000). Prior to this time, modern churches were not subject to any specific regulations other than basic health and safety requirements applicable to all built structures. Religious worship was a taken-for-granted aspect of cultural life that, by and large, went unquestioned and unregulated (Ponce Solé and Cabanillas 2011).

This state of affairs started to change as Spain's religious landscape began to diversify during the late 1980s and early 1990s, not just as a consequence of foreign immigration, but also due to the rise of new religious movements among the autochthonous population. Indeed, the first significant legal cases regarding the regulation of places of worship that came before national and regional courts were related to Pentecostal churches serving internal minorities, most notably Spanish gitanos (Roma). These cases dealt with the right of municipal governments to impose restrictions on places of worship to protect local residents from externalities derived from religious activities (e.g., problems related to noise and crowding) (Otaduy 2013). The cases forced the Spanish judiciary to adjudicate the limits of religious freedom vis-à-vis issues of public order and urban planning.

Some early rulings came down against municipal governments trying to impose various regulatory requirements on places of worship. For instance, a 1992 Supreme Court ruling prevented Madrid's city government from closing down a Pentecostal church that had elicited complaints related to noise. The city defended its action on the grounds that the church failed to obtain an appropriate building license and violated urban regulations pertaining to "disturbing, insalubrious, damaging and dangerous activities" and "public spectacles and recreational activities"[1]. The Supreme Court ruled that the license and regulations in question were meant for industrial or commercial enterprises, and that they constituted an unwarranted obstacle to the right of religious worship. The ruling specified that significant evidence of disturbance to public order needed to be demonstrated in order to limit religious worship, and that in the absence of such evidence, rulings should be made in accordance with the principle of favor libertatis (in favor of freedom), so as not to unduly infringe upon a fundamental right (Otaduy 2013).

Toward the turn of the 21st century, however, regional courts increasingly ruled in favor of municipalities seeking to regulate places of worship in accordance with regulations and ordinances they deemed appropriate. Such rulings were issued during a period when religious diversity was rising substantially as a result of foreign migration. The large-scale arrival of immigrants from South America, Africa, Asia, and Eastern Europe led to the proliferation of churches, mosques, and other temples catering to religious minorities, especially Evangelical Protestants, Muslims, and Orthodox Christians. Data on minority places of worship in Catalonia were first collected systematically in 2004, at which point 722 minority temples were identified. By 2007, this number had reached 915, and there are now over 1400 minority places of worship in the region (Directorate General of Religious Affairs 2019).

As places of worship began to rise in number during the 2000s, local conflicts surrounding their presence or proposed establishment became increasingly common. Mosques in particular became a major source of controversy, especially in Catalonia. The first major anti-mosque campaign to gain widespread media coverage, both locally and nationally, was organized in the Catalan municipality of Premià de Mar. Although the controversy began during the mid-1990s, it intensified significantly between 2001 and 2002, and served as a launching point for the far-right party "Coalition for Catalonia" (Plataforma per Catalunya), which subsequently gained political representation in several Catalan municipalities. Other anti-mosque campaigns were also initiated during this period in Banyoles (1999), Reus (2000), Granollers (2000, 2001), Lleida (2001), and Mataró (2001). These campaigns, it should be noted, surfaced prior to the terrorist attacks of 9/11 in New York City. They were followed by anti-mosque mobilizations in Torroella de Montgrí (2001), Badalona (2002, 2005–2007), Figueres (2002), Viladecans (2002), Les Franqueses del Vallès (2003), Barcelona (2004), Llagostera (2004), Mollet del Vallès (2004), Sant Feliu de Guixols (2004), Santa Coloma de Gramenet (2004), and Vilafranca del Penedès (2004). Like the anti-mosque campaign in Premià de Mar, the campaigns organized in Badalona and Santa Coloma de Gramenet involved large street demonstrations and garnered significant media coverage. Between 1995 and 2009—the year the LCW was enacted—29 of the 50 municipalities where anti-mosques campaigns materialized were located in Catalonia.

I have written extensively elsewhere about the sources of hostility toward mosques Catalonia (Astor 2012, 2016, 2017). To summarize, in addition to the fact that Catalonia is home to Spain's largest Muslim population, Muslims in Catalonia—especially in the greater metropolitan area of Barcelona—tend to be spatially concentrated in neighborhoods of post-industrial municipalities inhabited by historically marginalized segments of the populace, most notably working-class internal migrants from other Spain regions and their offspring, as well as gitanos. Many of these neighborhoods have a history of grassroots struggle dating back to the "neighborhood movement" of the 1960s and 1970s (Marín Corbera 2004). However, what began as a species of "militant particularism" (Harvey 1995) that connected urban demands to larger aspirations of universal emancipation and democracy during Franco's dictatorship, gradually transformed into a "defensive localism" as the movement unraveled during the 1980s and 1990s. The ongoing sense of abandonment and neglect pervading many of these neighborhoods has contributed to the integration of disputes surrounding mosques into broader struggles over urban justice and social position that preceded the large-scale arrival of Muslim immigrants from Africa and South Asia to Catalonia during the 1990s and 2000s.

The proliferation of anti-mosque campaigns in Catalonia generated heightened pressure for municipal authorities to develop clearer regulations for the establishment of places of worship. Nevertheless, determining which regulations were applicable to places of religious gathering was often a difficult task, as it required their reclassification into non-religious categories that rendered them legible within existing urban regulatory frameworks (Astor and Griera 2016). A range of actors, including municipal officials, urban planners, and experts in ecclesiastical and urban law, participated in determining which regulatory requirements were suitable for places of worship. Their determinations varied

significantly depending on the municipality in question (Martín García 2012), and they often appeared to be driven more by popular pressure than by strictly legal considerations. Several local authorities complained that they had an excess of discretion and lacked a precise legal basis for justifying their regulatory decisions. Pressure from these authorities played a key role in generating impetus for the development of a legal initiative at the regional level.

## 4. "Killing Two Birds with One Stone": The Pragmatic and Political Dimensions of the LCW

Discussions about drafting the LCW emerged during the mid-2000s. As controversies over mosques were receiving significant media coverage and eliciting concern among an increasing number of public officials, the Catalan independence movement was gaining momentum due to heightened tensions between Spain and Catalonia. In 2006, a new Statute of Autonomy affirming Catalonia's distinctiveness as a nation and expanding its powers of self-government was approved by the Spanish Congress and supported by a popular referendum. With respect to religious affairs, Article 161 states that the Generalitat (Catalonia's regional government) has "exclusive competency over matters of religious entities that carry out their activities in Catalonia", as well as "executive competency in matters related to religious freedom" (Boletín Oficial del Estado 2006, p. 27298).

The Statute faced immediate and intense opposition from right-wing and centralist political parties and elites. The PP's Parliamentary Group brought a case to Spain's Constitutional Court citing numerous articles of the Statute as unconstitutional, generating significant outrage among a large segment of the Catalan populace and fomenting nationalist sentiment in the region (Blanke and Abdelrehim 2015)[2]. Consequently, Catalan nationalist parties became more assertive in their struggle for political autonomy, culminating in the organization of a series of non-binding popular referendums on Catalan independence. Politics in the region came to revolve increasingly around questions of autonomy and independence, as opposed to more traditional Left–Right divisions.

The confluence of heightened concerns regarding religious diversification due to the proliferation of anti-mosque campaigns and rising nationalist sentiment stemming from contention over the new Statute of Autonomy is crucial for understanding the motivations driving the drafting of the LCW. During the mid-2000s, Catalonia's Directorate General of Religious Affairs (DGAR), a regional agency dedicated to addressing religious issues, was administered by the Republican Left Party (ERC), as it controlled the Vice Presidency in a coalition government with the Catalan Socialist Party and the Catalan Green Party. The ERC, a pro-independence party that has long sought to promote Catalonia's political autonomy and to affirm its cultural distinctiveness as a nation, consequently took a leading role in developing the LCW. By creating a new law dedicated exclusively to the regulation of places of worship and framing it as a distinctively Catalan law, the ERC's leadership sought simultaneously to respond to the problems resulting from a lack of clear guidelines for regulating mosques and other temples serving religious minorities, to enhance Catalonia's institutional autonomy vis-à-vis Spain by consolidating its competency in the area of religious affairs, and to symbolically affirm Catalonia's political and cultural distinctiveness by elaborating a framework for religious governance that reflected Catalan values and principles.

The nationalistic aspects of the LCW were evident in the symbolic rhetoric employed by Josep Lluís Carod Rovira, the leader of the ERC and Vice President of the Generalitat, during his initial presentation of the law to the Catalan Parliament in 2008. He touted the law as a "pioneering project, the first [law] in Europe that specifically regulates the conditions of centers of worship". He added that Catalonia was once again demonstrating itself to be "an advanced country" that opened up new avenues for legal innovation and that set an example for other Spanish regions (Diari de Sessions del Parlament de Catalunya 2008, p. 4). The proposed law, he contended, was a "courageous project" that filled a legal vacuum and responded to contemporary challenges related to religious diversity that had hitherto been ignored by the Spanish State. He emphasized how the LCW embodied the

values of "positive secularism" (laïcitat positiva), which included respect for all religions, solidarity, justice, and social cohesion. He concluded by praising the inclusive, plural, and democratic process by which the law was drafted, mentioning the input received from diverse entities, including the League for Secularism, Catalonia's Catholic bishoprics, and organizations representing Protestants, Jews, Muslims, and other religious minorities [3]. Implicit in Carod Rovira's remarks was an affirmation of Catalonia's distinctiveness vis-à-vis Spain with respect to its approach to meeting the challenges posed by religious pluralism. His portrayal of Catalonia as a pioneering and advanced country signaled its divergence from a more traditional and conservative Spain that was less inclusive and less accommodating of religious difference. Spain's long history of religious oppression and intolerance of diversity thus served as an implicit backdrop and resource for explaining and justifying the development of a uniquely Catalan law for regulating religious diversity.

## 5. Secularism a la Catalana

Legislative preambles are the "the symbolic site par excellence" for the articulation of identitarian dimensions of law (cf. Zubrzycki 2001, p. 633). The LCW's preamble includes a detailed elaboration of the aspects of the 2006 Statute of Autonomy and other existing legislation that establish Catalonia's competences in the areas of religious affairs and urban planning. The concluding paragraph elaborates the normative principles underpinning the law:

> In this way, the present law, on the basis of laïcitat—that is, on the basis of respect for all religious options and [options] of thought, and their values, as an integrative principle and common framework for coexistence, aims to regulate centers of worship under terms of neutrality and with the exclusive end of facilitating the practice of worship and preserving the safety and health of the premises and the fundamental rights of all citizens relative to public order. In this way, and on the basis of collaboration, it aims to strengthen the values that already characterize the common space of our society: coexistence, respect for plurality, equality of democratic rights, and the responsibility of the citizenry, without any kind of discrimination, [to participate] in the national construction of Catalonia. [4]

The bulk of this paragraph does not, on its face, seem overly controversial. After all, it is not especially partisan to invoke the principles of religious freedom and worship, health and safety, peaceful coexistence, and democratic rights. However, the preamble's inclusion of the symbolically charged term "laïcitat" and its reference to the "national construction of Catalonia" generated alarm among conservative and centralist political elites. During the initial parliamentary debates, José Domingo of the conservative Citizen's Party (Ciudadanos) condemned the authors of the LCW for needlessly linking religious issues with matters of national identity:

> It turns out that the values of Catalan society are now the national construction of Catalonia, as it figures in the presentation of [the law's] motives. Please! Are you speaking about religion? By the way, I recall there is one [religion] which says that our kingdom is not of this world, and you go and focus on the national construction of Catalonia in religious matters. Come on! Perhaps this is a little excessive, no? Control yourselves. We ask that you control yourselves. If you speak of religion, speak of religion, but don't talk about so many things at once
>
> (Diari de Sessions del Parlament de Catalunya 2008, p. 17).

In contrast to the representatives of other parties, Domingo made a point of speaking in Spanish rather than Catalan to signal his anti-nationalist posture. He argued that the regulation of religious buildings should be dealt with at the national or local levels, and not by the regional government. He added that problems of interreligious coexistence (i.e., mosque conflicts) should be addressed through pedagogy rather than new laws.

During the parliamentary debates preceding the LCW's enactment in 2009, María Ángeles Olano of the conservative and anti-nationalist Popular Party of Catalonia (PPC) took direct aim at the inclusion of laïcitat in the LCW's preamble:

> We consider that the position the government has taken, above all in its preamble, is a grave error. The preamble that has been established is an unacceptable affirmation and contrary to Article 16 of the Constitution, as it attempts to regulate centers of worship on the basis of laïcitat. The Spanish State is a non-confessional state; it is not a state of secular confession. This intended definition goes against religious liberty; religion is not a stigma to eliminate, and therefore it is important to recognize its presence. The State cannot ignore the existence of religion in society, it cannot situate it at the margins, as laïcisme (militant secularism) pretends to do, by excluding the religious dimension from the public sphere
>
> (Diari de Sessions del Parlament de Catalunya 2009, p. 6).

In order to understand Ángeles Olano's vehement criticism of including the term laïcitat in the LCW's preamble, as well as the broader controversy that emerged surrounding the law's normative grounding, it is necessary to first provide some background regarding the existing framework for regulating church-state relations in Spain and the manner in which it has been challenged by the Catalan Left. The current Spanish Constitution was drafted in the aftermath of Francisco Franco's death, which brought an end to 36 years of dictatorship and to the regime's ideological lynchpin of National Catholicism. In an effort to evade societal polarization and conflict, political elites agreed to put aside their deep-seated animosities and desires for retribution so as to negotiate a constitution that reflected a general orientation toward compromise and consensus. The longstanding historical cleavages surrounding religion in Spain made the process of designing a new framework for church-state relations a delicate matter. Given the need to find a formula that was acceptable to elites on the Right, as well as the Left, a militantly secularist constitution akin to that of the Second Republic (1931–1939) was out of the question (Gunther and Blough 1981). An agreement was eventually reached to disestablish the Catholic Church, but to include an article (16.3) specifying that the State must develop cooperative relations with "the Catholic Church and other confessions"[5]. This article constituted the bedrock of Spain's 'cooperative' framework for regulating church-state relations, and the explicit mentioning of the Catholic Church signaled recognition of its special status in Spanish society (Motilla de la Calle 1985).

The State's relations with the Catholic Church have been governed by a series of international agreements with the Vatican established in 1976 and 1979. These agreements preserved several of the Catholic Church's privileges in the areas of finance, education, and public patrimony[6]. They served as a template for subsequent agreements established with Jewish, Muslim, and Protestant federations in 1992. The 1992 agreements were made possible by the prior recognition of these three religions as "deeply rooted" (de notorio arraigo) in Spanish society, a process outlined in the 1980 Organic Law on Religious Liberty (LOLR) (Fernández-Coronado 1995). The State subsequently granted recognition to The Church of Jesus Christ of Latter-day Saints (2003), Jehovah's Witnesses (2006), Buddhism (2007), and Christian Orthodoxy (2010). In principle, the federations representing these religions are therefore eligible to sign cooperative agreements with the State, though none have done so as of yet. Spain's church-state regime may thus be characterized as a 'graduated model of religious recognition and cooperation' in which the explicit mentioning of the Catholic Church in the Constitution and the State's international agreements with the Vatican represent the highest form of recognition and cooperation, followed by the 1992 agreements with Jewish, Muslim, and Protestant federations, and the lesser forms of recognition granted to other confessions.

The ERC and other elements of the Catalan Left have long been critical of the framework for regulating church-state relations established by the Constitution and the LOLR due to its privileging of the Catholic Church and its 'failure' to create a stricter separation between church and state. The ERC and the Catalan Green Party have thus proposed

several national-level initiatives aiming to create a clearer separation between church and state, to limit the presence of religion in the public sphere, and to reduce the Catholic Church's financial and symbolic privileges (e.g., by prohibiting religious organizations from receiving direct public subsidies or by proposing restrictions on the participation of religious leaders in state events and ceremonies) (Astor 2020).

Given the ERC's historical position on matters of church and state, the inclusion of laïcitat in the LCW's preamble was interpreted by conservatives as having broader symbolic signification that challenged the core principles underlying Spain's existing framework for church-state relations. Conservative politicians and Church officials leveled two main criticisms of the LCW's general framing. The first related to its implicit grounding in a militant form of secularism hostile to religion. The second related to its treatment of Catholic churches and the threat it posed to the status of Catholicism in Spain and Catalonia.

With respect to the first criticism, the aforementioned statement by the PPC's Ángeles Olano was illustrative of the concerns among conservatives emanating from the ERC's known secularist positioning on religious issues. Although the definition of laïcitat included in the preamble was quite vacuous and far from militant, the term itself was associated with a French approach to regulating religious affairs perceived as aggressively secularist and hostile to religion[7]. Bowen (2007) has highlighted how, even in France, laïcité "is one of those 'essentially contested concepts' that is politically useful precisely because it has no agreed-on definition". As is evident from the debate over the LCW's preamble, the principle of laïcitat is similarly polysemous, malleable, and contested in Catalonia. In contrast to France, however, the term laïcitat (or laicidad in Spanish) does not have broad bipartisan appeal, and there is no general consensus that it should serve as a normative basis for church-state relations and religious governance. Including the term laïcitat in the LCW's preamble was viewed by the Right as potentially establishing a precedent that would permit the principle to gain a foothold in Spanish law, thereby pushing it in a more secularist direction[8].

In regard to the second criticism, a key challenge posed by the LCW to the graduated model of religious recognition and cooperation was its uniform application to all places of worship. As mentioned above, Catholic churches historically had been, by and large, exempted from municipal regulations and ordinances as a result of their taken-for-granted status as part of the socio-cultural and material landscape of Spanish towns and villages. Catholic officials expressed concern that the law might be applied retroactively to old church buildings that would have difficulty complying with its requirements. As a means of addressing such concerns, a provision was added to the LCW that exempted religious buildings included in Catalonia's registry of cultural patrimony, essentially all of which were Catholic. It was thus highly unlikely that the law would be applied retroactively to old churches.

The misgivings among Church officials and conservative political elites, however, had less to do with the LCW's potential impact on actual churches than with the symbolic implications of subjecting Catholic structures to the same regulations as minority places of worship. The specification of laïcitat as the ideological foundation for the LCW, as well as a core element of Catalan history and identity, was perceived as contributing to the valorization of a form of 'secular heritage' with respect to which all religions had equal standing. The uniform application of the law to all buildings designated for religious gathering was perceived as undercutting Catholicism's special status in Catalonia vis-à-vis other religions.

Joan Enric Vives, the secretary of the Tarragona Episcopal Conference and spokesman for Catalan bishops, criticized the draft bill for using "the same law to regulate the very diverse realities of churches, synagogues, mosques and other centers" (Noguer 2007). The following year, he clarified the Church's position on the initiative, stating, "It will probably always be inappropriate to use the same legislation to regulate centers of worship and gathering that vary so much by religious confession and that have such an asymmetric

presence in Catalan society" (Guil 2008). The archbishop of Barcelona, Lluís Martínez Sistrach, likewise criticized the law for treating all confessions the same despite "very substantial differences and specificities" (Llaquet de Entrambasaguas 2013, p. 52).

The general position of Catholic authorities in Catalonia and elsewhere in Spain has been that 'equity is not synonymous with equality or uniformity', given the importance of Catholicism to Spanish identity, history, and culture[9]. This slogan was foregrounded in a document entitled "Religion in the Catalonia of the Future", which was presented to the Catalan Parliament in 2017 by the Catalan Union of the Religious, a Barcelona-based ecclesiastical entity. In explaining the slogan's meaning, the authors wrote, "Diverse religions have different religious and cultural characteristics and needs; they have diverse histories, implantation, rootedness and representativeness".

Conservative political elites voiced similar concerns regarding the LCW's failure to give sufficient recognition to the historical centrality of Catholicism to national identity and heritage in Spain and Catalonia. During the parliamentary debates over the law in 2008, Ángeles Olano argued that the LCW jeopardized the "Catholic roots not only of Catalonia, but also of Spain" (Diari de Sessions del Parlament de Catalunya 2008, p. 7). Maria Glòria Renom of the center-right nationalist party Convergence and Union (CiU) stated that, although all religions deserved respect and support, "it is also necessary to observe that the sociological and cultural weight that each of these has in Catalonia is distinct. Religious freedom, thus, means respecting minorities but keeping in mind the majority and the historical and traditional values that have constituted catalanidad (Catalanness)" (Diari de Sessions del Parlament de Catalunya 2008, p. 11). The question of whether Catholic organizations should receive special exemptions and benefits vis-à-vis other religious organizations due to Catholicism's importance to Spanish history, identity, and culture remains central to the symbolic struggles over church-state relations in Spain.

## 6. The Passage and Aftermath of the LCW

The LCW was formally passed in 2009 by a vote of 119 to 13, with 1 abstention. The only party that remained steadfast in its opposition to the law was the PPC. CiU's eventual decision to support the LCW was likely connected to the law's nationalist and pro-independence undertones, as well as CiU's knowledge that it had a strong chance of winning the regional elections the following year, in which case it would have the opportunity to amend the law as it saw fit. The implementation of the LCW was contingent upon the formulation and passage of an additional regulation that would clarify several of the technical requirements included in the law, such as those pertaining to safety and hygiene. This regulation was passed in July of 2010, enabling the LCW to be put into effect.

Four months after the LCW was implemented, CiU won Catalonia's general election. The following year, the party proposed several modifications, including the inclusion of a provision stating that licensing decisions should be made in accordance with "the degree of implantation and rootedness of each of the churches, confessions and religious communities" (Butlletí Oficial del Parlament de Catalunya 2011, p. 22). CiU also proposed the addition of a clause stating that the technical requirements of the LCW should "respect the architectonic, cultural, traditional, and historical characteristics, and the artistic elements of centers of worship that already exist" (Butlletí Oficial del Parlament de Catalunya 2011, p. 22). The main aims of these clauses were to symbolically affirm the special status of Catholicism in Catalonia and to shield Catholic churches from overzealous local authorities who might demand costly renovations of older church buildings. The proposed modifications, however, never reached a vote[10].

In 2012, the Basque Socialist Party drafted a law on centers of worship that drew clear inspiration from the Catalan LCW. The law had been developed in the aftermath of a heated dispute over a mosque in Bilbao and the consequent passage of restrictive legislation pertaining to places of worship in the city (Ruiz Vieytez 2014). However, the proposal failed to come to fruition. Several years later, the proposal was resuscitated under the name, "Law on Places, Centers of Worship and Religious Diversity in the Autonomous

Community of the Basque Country". The law was eventually presented before the Basque Parliament in 2019, but it has yet to arrive at a vote.

A simple interpretation of the Basque proposal would be that, given the Basque Country's status as a stateless nation, the nationalist politics at play in debates over religious governance are comparable to those in Catalonia. There are indeed many similarities between the Basque and Catalan laws. The Basque law makes specific mention of "positive secularism" as its foundational principle, includes an article requiring municipal governments to ensure adequate space for religious buildings in their zoning schemes, and details a series of health and safety requirements that resemble those elaborated in the Catalan LCW. However, there is no mention of Basque identity, and the presentation of the law to the Basque Parliament did not include symbolic references celebrating the Basque nation akin to those made by the ERC's Carod Rovira when introducing the LCW to the Catalan Parliament.

Here, it is important to underline that the pro-independence movement in the Basque Country is weaker than in Catalonia, and the coalition government which proposed the most recent incarnation of the law on places of worship to the Basque Parliament was led by the conservative and Christian democratic Basque Nationalist Party (PNV). The PNV is not at all comparable to the ERC and other elements of the Catalan Left with respect to its position on church-state relations. If there are overtly nationalistic or secularist undertones present in the law, they are much less apparent or central, which explains why the law has not elicited significant criticism from Catholic officials or the PP.

This highlights how the nationalistic aspects of a given law, in this case the Catalan LCW, do not necessarily carry over to other laws for which it serves as a model. With the enactment of the Catalan LCW, regional laws regulating places of worship became part of the general policy repertoire available for consideration by other regional administrations when determining how to deal with practical dilemmas of governance. In the case of the Basque Country, the conflict over places of worship in Bilbao led to the formation of a committee of experts to analyze the issue and develop possible solutions at the regional level (Ruiz Vieytez 2014). The Catalan LCW provided the most obvious and straightforward policy option, though its nationalistic dimensions were not the main source of its appeal.

The enactment of the Catalan LCW and concerns generated by the potential proliferation of the Catalan model to other regions like the Basque Country led Spain's central government to develop national-level legislation related to places of worship as part of a "Law of Local Administration Rationalization and Sustainability", also known as the "Local Reform Law", which was passed in 2013[11]. The article of the law pertaining to places of worship stipulated that only urban licenses, and not activity licenses, were required for establishing places of worship, so long as the activities realized were communicated to the appropriate authorities. This was a direct challenge to the Catalan LCW, which had created a new activity license for the express purpose of regulating places of worship[12].

In the context of growing tension between Spain and Catalonia over issues of political autonomy and independence, the Local Reform Law was perceived as a threat to Catalonia's rights of self-government. The following year, the Catalan Parliament filed a motion with the Spanish Supreme Court contesting the Local Reform Law's invasion of its competencies in religious matters, as well as in other social domains. The LCW remains the law of the land in Catalonia, despite the central government's efforts to undermine its raison d'être. In the decade that has passed since its implementation, there have thus far been no significant legal battles over its application.

### 7. The LCW in Comparative Perspective: The So-Called "Anti-Mosque" Laws in Italy

As discussed above, the nationalistic aspects of the LCW revolved largely around efforts to signal Catalonia's distinctiveness from Spain, in part by highlighting its modern and inclusive approach to regulating religious diversity and accommodating the needs of religious minorities. Symbolic politics (Gusfield 1963) likewise figured centrally in recent legislation regulating places of worship in the northern Italian regions of Lombardy

(January of 2015), Veneto (April of 2016), and Liguria (September of 2016)[13]. In contrast to the Catalan LCW, however, the symbolic messaging implicit in these legislative initiatives was exclusionary and hostile to religious minorities, especially Muslims. The laws were proposed by right-wing coalitions that included representatives (or former representatives) of the Northern League, a far-right populist party whose anti-Muslim rhetoric has been a central part of its political program.

The authors of the Italian laws were explicit in promoting them as "anti-mosque laws" (Lo Giacco 2019), a framing meant to reassure the populace of their intention to protect them against the "Muslim threat". Whereas the Catalan LCW was an entirely new law created for the express purpose of regulating places of worship, the Italian regulations comprised a series of modifications to existing laws on urban planning and development. The initiatives were designed to impede the construction of new mosques by subjecting them to an assortment of bureaucratic obstacles, as well as by giving greater voice and influence to those opposing their establishment.

Given that explicitly discriminating against a particular religious minority would most certainly have been deemed unconstitutional, the authors of the Italian initiatives included provisions that subtly targeted mosques without referencing them explicitly. For instance, the Lombardy regulation differentiated between religious buildings associated with the Catholic Church and other religions whose respective representatives had signed agreements with the State, on the one hand, and religious buildings associated with religions lacking such agreements, on the other. The regulation stipulated that religions lacking an agreement with the State had to have a "widespread, organized, and stable presence" within the jurisdiction in question in order to benefit from full legal consideration. Given that Islam was the only major religion in Italy that had no formal agreement with the State, it is clear that this differentiation was made with the intention of targeting mosques. Municipal administrations would, in principle, be able to deny mosques authorization on the grounds that the presence of Muslims in a given jurisdiction was not sufficiently large or stable, even if the mosque satisfied other legal requirements. Since the requirements of size, organization, and stability were left unspecified, the regulation introduced a measure of discretion that could be exploited for discriminatory purposes (Chiodelli and Moroni 2017). This, along with a provision requiring the installation of surveillance cameras whose recordings would be accessible by the police, were two of the provisions found most objectionable by the Italian Constitutional Court. Consequently, they were left out of the Veneto and Liguria regulations, which largely mirrored the Lombardy regulation.

Another measure that indirectly targeted mosques in the Lombardy and Liguria laws was a provision stating that the architectural design of religious buildings had to be congruent with the "general and particular features of the landscape" in each region. This would make it highly difficult to construct mosques with minarets or other visibly Islamic characteristics, as they could easily be interpreted as clashing with their surrounding landscape. This provision was distinct from the modification to the Catalan LCW dealing with architectonic and cultural matters proposed by CiU, insofar as CiU's proposed modification was mainly intended to protect the structural integrity of Catholic churches subjected to the exigencies of the LCW, and did not specifically target mosques in any way. The provision on architectural conformity included in the Lombardy and Liguria laws, by contrast, had the clear intention of limiting the visible presence of 'alien' religious structures (Mocchi 2018).

The Veneto regulation contained an additional provision requiring that non-religious activities in places of worship be conducted in Italian, highlighting the assimilationist agenda of its authors. It also introduced zoning restrictions that would, in effect, relegate new religious buildings to urban peripheries. Arguably the most controversial element of each of the three regulations was a provision enabling local governments to hold consultative referenda on whether or not to permit the construction of religious buildings in their respective jurisdictions. As a consequence, mosques would not only have to meet all legal requirements, but would also have to be supported by the general public to receive authorization. In contrast to the LCW, which sought to limit the impact of social pressure

and the discretion of municipal authorities in deciding whether to authorize places of worship, the Italian regulations aimed to achieve just the opposite.

## 8. Conclusions

Over the course of this article, I have highlighted the importance of historical timing and conjunctural causality for explaining the decision of Catalan policymakers to pursue a legally innovative solution to regulating places of worship that broke with standard European approaches. The LCW was designed to promote the dual aims of pragmatically addressing dilemmas regarding the regulation of minority religious temples, on the one hand, and of institutionally codifying and symbolically affirming Catalonia's political autonomy and cultural distinctiveness, on the other. The particular manner in which religious governance and nationalist politics became entwined in the LCW resulted from the temporal confluence of heightened contention over mosques and rising nationalist sentiment due to the inflaming of tensions between Spain and Catalonia surrounding the new Statute of Autonomy.

When analyzing the nexus between religion, law, and politics, it is thus crucial to attend not only to entrenched structural or cultural features of the sites we study, but also to contingent conjunctures of events that shape the dynamic interplay between the three spheres. Had anti-mosque campaigns in Catalonia become sufficiently alarming to spark a regional legal response prior to the rise in nationalist sentiment and mobilization, the Generalitat would likely have opted for a legal solution that was more in line with precedents set elsewhere in Europe. In other words, Catalan nationness, and not just the more constant and generic features of Catalan nationalism or nationhood, is essential for explaining the development and enactment of the LCW. It remains to be seen what effect the precedent set by the Catalan LCW will have on the future regulation of places of worship elsewhere in Spain or in Europe. The Basque example illustrates the potential influence of the LCW as a new policy option, even in contexts that are less nationalistically mobilized, or where the parties in power have little interest in the ideologically charged features of the law.

**Funding:** This research received no external funding.

**Institutional Review Board Statement:** Not applicable.

**Informed Consent Statement:** Not applicable.

**Data Availability Statement:** Not applicable.

**Conflicts of Interest:** The author declares no conflict of interest as author of this article.

## Notes

[1] The specific regulations in question were the 1961 "Regulation on Disturbing, Insalubrious, Damaging and Dangerous Activities" (Reglamento de actividades molestas, insalubres, nocivas y peligrosas) and the 1982 "Regulation on the Policing of Public Spectacles and Recreational Activities" (Reglamento de policía de espectáculos públicos y actividades recreativas). The laws may be consulted at https://www.boe.es (accessed on 26 January 2021).

[2] In 2010, the Constitutional Court ultimately ruled that 14 of the Statue's articles were unconstitutional, and that another 27 had to be reinterpreted in accordance with the Constitution. The court also declared references to Catalonia's status as a nation in the preamble as lacking legal force (Blanke and Abdelrehim 2015).

[3] Llaquet de Entrambasaguas (2013) provides a detailed description of the input from Muslims and other religious minorities at each stage of the LCW's development.

[4] The full text of the law may be found at: https://www.boe.es/eli/es-ct/l/2009/07/22/16/con (accessed on 25 January 2021).

[5] See Article 16 of the 1978 Constitution: https://www.boe.es/legislacion/documentos/ConstitucionINGLES.pdf (accessed on 25 January 2021).

[6] The agreements stipulated the gradual phasing out of direct State financing of the Catholic Church. In practice, however, the Church continues to receive substantial public subsidies.

7    In Spanish, a distinction is often drawn between '*laicidad*' (moderate secularity) and '*laicismo*' (radical secularity) (see Ramírez García (2012)). Conservatives and others on the Right commonly accuse the Left of pursuing an agenda that is radically secular and grounded in a vision of religion that sees it as a danger to public life.

8    The concept of '*laicidad positiva*' (positive secularity) was mentioned in a 2001 Constitutional Court Ruling affirming the right of the Unification Church to gain recognition as a religious organization and to register with the Ministry of Justice. Prior to the LCW, however, the principle of secularity had not been included in actual legislation.

9    The document may be viewed at: http://bisbatsolsona.cat/wp-content/uploads/2017/03/El-fet-religi%C3%B3s-en-la-Catalunya-del-futur.pdf (accessed on 25 January 2021).

10   In 2014, a measure was passed extending the timeframe in which places of worship were required to comply with the LCW's technical requirements from five to ten years. This measure was connected to the passage of a law entitled, the "Law of Fiscal, Administrative, Financial and Public Sector Measures" (Llei 2/2014, del 27 de gener, de mesures fiscals, administratives, financeres i del sector públic).

11   For the full text of the law, see: https://www.boe.es/diario_boe/txt.php?id=BOE-A-2013-13756 (accessed on 25 January 2021).

12   The article also required that religious communities establishing places of worship be registered with the National State Registry of Religious Entities.

13   The Lombardy initiative may be viewed at: http://normelombardia.consiglio.regione.lombardia.it/NormeLombardia/Accessibile/main.aspx?view=showdoc&iddoc=lr002015020300002 (accessed on 25 January 2021). The Veneto initiative may be viewed at: http://www.consiglioveneto.it/crvportal/leggi/2004/04lr0011.html (accessed on 25 January 2021). The Liguria initiative may be viewed at: http://lrv.regione.liguria.it/liguriass_prod/articolo?urndoc=urn:nir:regione.liguria:legge:2016-10-04;23&dl_t=text/xml&dl_a=y&dl_id=&pr=idx,0;artic,0;articparziale,1&anc=art2 (accessed on 25 January 2021).

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
