# Peer review of "Nationalist Mobilization, Ethno-Religious Contention, and Legal Innovation in a Stateless Nation: Explaining Catalonia’s 2009 “Law on Centers of Worship”"

_religions, doi:10.3390/rel12050295_

Round 1
Reviewer 1 Report
This is an original, well-researched, and well-written article. Congratulations to the author.
Author Response
Thank you for your kind assessment.
Reviewer 2 Report
The paper offers an interesting discussion on Catalonia's Law on Centers of Worship considering legal, political and socio-religious processes and debates that accompany the enactment of this Law. The author's perspective to analyze the law-making in the context of a broader nation-building process contributes to existing theories of religious freedom. I have some minimal comments to the author:
- The concept of stateless nation requires some definition with references to the research conducted on that topic. In this regard, lines 54-56 require references relevant to the theoretical argument.
- Lines 131-133 - the sentence is not complete (after "religious"...is missing the noun).
- Lines 267-269 - the conclusion after the comparison is too brief, please elaborate it.
- It is beneficial if the author puts theoretical references to the work of A. Sarkissian (2015) who explained the role of religious divisions in the past and state religion governance for the process of nation-building.
Author Response
Thanks for these helpful comments. As you suggested, I have included a definition of 'stateless nation' which I borrow from Guibernau (lines 56-60). I have also fixed the mistake on lines 131-133 and elaborated the conclusion to the section you reference (lines 274-276). I looked at A. Sarkissian's 2015 book, but it focuses on religious oppression in authoritarian societies, and I did not see the immediate relevance.
Reviewer 3 Report
I find that this paper is deeply and strong reasoned and facilitate a meaningful discussion. It presents a rethinking of the question of Catalonia nationalism related with the religious question and it offers valuable information and an original academic research. I have no objections.
I only can add some suggestions:
- On p. 3, I will arise the question if the nationalist movement finds in these minorities an enhacement since they can also reach more voters through them?
- On p. 11, second paragraph when it is mentioned the Basque Law by comparison with the LCW, at the end of the paragragh It could be interesting to include any particular examples in one Law and in the other.
- On the same page it is referred to the Local Reform Law, but I miss more information about this Law and the date of its enactment.
-I find that the conclusion might be have a lack of argumentation to the claim from which we understand that this Law of regarding the regulation of minority religious temples was a primary neccesity and the nationalistic issue did an "utilization" from the nationalism and no a Law made purposely.
-Maybe it will be a great addition to explore two more points or in a further research.:
- Apart from the preamble of the LCW and Carold Rovira defence, has been other statements in the direction of the thesis of the article?
- In the last elections has been claims of the interest of some of the nationalistic parties to attack the vote of the nationalized immigrants in favor of the independence movement. There has been any conection to this law?
With regard to Bibliography, there are two articles which it may be also worth to view. These are:
Martín García, M. del Mar, "Derecho de libertad religiosa y establecimiento de centros de culto. A propósito de su desarrollo legal en Cataluña", Revista Española de Derecho Constitucional, núm. 94, enero-abril (2012), pp. 239-265.
Ramírez García, Hugo, "Derecho y religión. Notas sobre la lectura contemporánea de la libertad religiosa en Europa", Boletín Mexicano de Derecho Comparado, vol. XLV (2012), núm. 133, 2012, pp. 283-315.
On another hand, I presume that the text format will be edit, but in any case, I found a lot words incorrectly cutted at the end of the lines. As well as It must justify the paragraphs.
Author Response
Thanks for these helpful comments. I think nationalist movements are trying to recruit minorities into their ranks, and I have discussed this in other publications. However, I do not see this as a primary motivation for this law in particular. As you suggested, I have added some more concrete details about how the Basque law compares to the Catalan law on places of worship (p. 11). With regard to the Local Reform Law, I include the year but I do not enter into detail regarding the specifics of the law as it is somewhat tangential to my main argument.
I'm not sure what you mean in your remarks about the conclusion. Something about utilisation and nationalism, but I can't make out the point you are raising. The law was purposive and instrumental for advancing a nationalist agenda.
The points you make about future research are good, but I do not necessarily think I should discuss them at length here. I do talk about Carod Rovira's initiatives regarding nationalism and religious diversity in other publications referenced in the paper.
Finally, I cite the Martín García and Ramírez García articles you mention at different points in the text and in a footnote (#7).